# FARSB Facilitates Hepatocellular Carcinoma Progression by Activating the mTORC1 Signaling Pathway

**DOI:** 10.3390/ijms242316709

**Published:** 2023-11-24

**Authors:** Yaofeng Wang, Gengqiao Wang, Shaobo Hu, Chuanzheng Yin, Peng Zhao, Xing Zhou, Shuyu Shao, Ran Liu, Wenjun Hu, Gang Logan Liu, Wenbo Ke, Zifang Song

**Affiliations:** 1Department of Hepatobiliary Surgery, Union Hospital, Tongji Medical College, Huazhong University of Science and Technology, Wuhan 430022, China; wangyf045@163.com (Y.W.); wanggengqiao@outlook.com (G.W.); hsb9999@126.com (S.H.); dryincz@hust.edu.cn (C.Y.); gdzsz123@163.com (P.Z.); xzhou2009@hust.edu.cn (X.Z.); m202275950@hust.edu.cn (S.S.); 15271890477@163.com (R.L.); 2School of Life Science and Technology, Huazhong University of Science and Technology, Wuhan 430074, China; hu_wenjun@hust.edu.cn (W.H.); loganliu@hust.edu.cn (G.L.L.)

**Keywords:** FARSB, hepatocellular carcinoma, mTORC1, Raptor, ferroptosis

## Abstract

Hepatocellular carcinoma (HCC) is a common malignant tumor with high mortality. Human phenylalanine tRNA synthetase (PheRS) comprises two α catalytic subunits encoded by the *FARSA* gene and two β regulatory subunits encoded by the *FARSB* gene. *FARSB* is a potential oncogene, but no experimental data show the relationship between FARSB and HCC progression. We found that the high expression of FARSB in liver cancer is closely related to patients’ low survival and poor prognosis. In liver cancer cells, the mRNA and protein expression levels of FARSB are increased and promote cell proliferation and migration. Mechanistically, FARSB activates the mTOR complex 1 (mTORC1) signaling pathway by binding to the component Raptor of the mTORC1 complex to play a role in promoting cancer. In addition, we found that FARSB can inhibit erastin-induced ferroptosis by regulating the mTOR signaling pathway, which may be another mechanism by which FARSB promotes HCC progression. In summary, FARSB promotes HCC progression and is associated with the poor prognosis of patients. FARSB is expected to be a biomarker for early screening and treatment of HCC.

## 1. Introduction

Hepatocellular carcinoma (HCC) is currently the third most common cause of cancer death worldwide, and its incidence is increasing yearly. About 830,000 people die of HCC every year [1]. Hepatocellular carcinoma is the most common type of liver cancer, accounting for about 75–85% of all primary liver cancer [2]. However, the etiology and carcinogenesis mechanism of HCC are still unclear. Currently, the treatment methods for HCC include surgical resection, ablation, transcatheter arterial chemoembolization (TACE), targeted therapy, liver transplantation, immune node inhibitors and tumor infiltrating lymphocytes [3]. However, these treatments are not ideal due to their high recurrence and metastasis rates. Therefore, exploring the molecular mechanism of HCC development is expected to provide reliable indicators for early screening of HCC. In addition, these molecules are expected to become therapeutic targets for HCC in the later stage.

Aminoacyl-tRNA synthases (ARSs) are key enzymes in protein translation, which can connect specific amino acids to their homologous tRNAs to ensure the accurate decoding of mammalian mRNA [4]. Studies have shown that ARS is closely related to developing autoimmune diseases [5], genetic diseases [6] and cancer in organisms [7] and it plays a non-canonical role besides protein translation. Human phenylalanine-tRNA synthetase (PheRS) belongs to the class IIc ARSs subfamily. Studies have shown that increased expression of PheRS can promote cell growth and proliferation [8], while the loss of mitochondrial PheRS expression is associated with developmental delays and seizures [9]. PheRS comprises four subunits encoded by *FARSA* and *FARSB* genes, including two catalytic α subunits and two regulatory β subunits. FARSB has identified a pseudogene located on chromosome [10]. *FARSB* gene mutations can cause neurodegenerative diseases and diffuse brain calcification [11]. Moreover, *FARSB* biallelic mutations can lead to cirrhosis [12]. Studies have shown that FARSB is associated with tumors, and *FARSB* may be a downstream gene of the critical tumor suppressor gene *IGFBP7* in colorectal cancer [13]. *FARSB* was also identified as a hub gene for tobacco-related methylation probes in lung adenocarcinoma [14]. Recent studies have shown that FARSB is associated with hypomethylation and immune cell infiltration [15]. However, the mechanism by which FARSB affects the progression of hepatocellular carcinoma is not yet precise.

Mammalian target of rapamycin (mTOR) is a serine/threonine kinase, usually assembled into a complex including mTORC1 and mTOR complex 2 (mTORC2). The mTORC1 signaling pathway controls cell growth and proliferation by sensing fluctuations in nutrients and growth factors under environmental conditions, mediating the balance between anabolism and catabolism [16]. The dysregulation of mTORC1 signaling is associated with the progression of various human cancers [17]. The mTORC1 signaling pathway is the primary tumor initiation pathway in HCC [18], which is up-regulated in 40–50% of HCCs and plays a vital role in the development and progression of HCC [19]. In ARSs, previous studies have shown that seryl-tRNA synthetase (SerRS) can act as an amino acid sensor to regulate cell biological behavior through the mTOR signaling pathway [20]. At the same time, isoleucyl-tRNA synthetase (IleRS2) can activate the Akt/mTOR signaling pathway to promote the progression of small cell lung cancer [21]. Therefore, this study confirmed the regulatory effect of FARSB on HCC progression through the mTORC1 signaling pathway.

Ferroptosis is an iron-dependent form of cell death driven by excessive lipid peroxidation, which is related to developing and treating various tumors [22]. The main manifestations of ferroptosis were Reactive Oxygen Species (ROS) accumulation, increased glutathione and decreased glutathione peroxidase 4 (GPX4) activity and content [23]. In ARS, only cysteyl-trna synthetase (CysRS) has been elucidated to be associated with ferroptosis. Knockdown of CysRS in pancreatic cancer cells can inhibit ferroptosis by activating the transsulfuration pathway [24]. Recent studies have shown that CysRS promotes ferroptosis in esophageal squamous cell carcinoma by regulating GPX4 expression [25]. In addition, activating the mTOR signaling pathway up-regulates Sterol-regulatory element binding proteins (SREBPs) to inhibit ferroptosis through stearoyl-CoA desaturase-1 (SCD1) [26]. This study verified that FARSB inhibits ferroptosis through the mTOR signaling pathway.

## 2. Result

### 2.1. FARSB Is Highly Expressed in Hepatocellular Carcinoma

Bioinformatics analysis was performed using the transcriptome sequencing data of patients in the The Cancer Genome Atlas (TCGA) database. Compared with adjacent tissues, the expression level of *FARSB* in hepatocellular carcinoma tissues was significantly up-regulated (Figure 1A,B). At the same time, we performed immunohistochemical staining analysis on postoperative tissue samples of liver cancer patients. FARSB protein levels were significantly up-regulated in liver cancer tissues compared to adjacent tissues (Figure 1C). In addition, Western blot and RT-qPCR analysis of LO2, Huh7, HepG2 and MHCC97H cells confirmed that the expression levels of FARSB mRNA (Figure 1D) and protein (Figure 1E) were up-regulated. Subsequently, the immunofluorescence intensity of FARSB in two hepatocellular carcinoma cell lines (Huh7 and MHCC97H) was significantly higher than that of LO2 (Figure 1F). The above results suggest that the expression of FARSB in hepatocellular carcinoma was significantly increased.

### 2.2. High Expression of FARSB Is Associated with Poor Prognosis of Patients

The clinical correlation analysis of FARSB was performed based on patients’ clinical information from the TCGA database. The high expression of FARSB was associated with tumor grade and T stage (Figure 2A). In addition, univariate and multivariate Cox regression analyses were performed based on clinical information such as patient gender and tumor distant metastasis. The analysis results were visualized by drawing forest plots. The results showed that FARSB may be an independent prognostic factor for patients with liver cancer (Figure 2B,C). Subsequent survival analysis of patients showed that the overall survival rate and disease-free survival rate of patients in the low expression group were significantly higher than those in the FARSB high expression group (Figure 2D), indicating that the high expression of the FARSB gene was closely related to the poor prognosis of patients.

### 2.3. FARSB Knockdown Suppresses the Proliferation and Migration of HCC Cells 

In order to verify the cancer-promoting effect of FARSB in HCC, the FARSB knockdown cell model was constructed by transfecting Huh7 and MHCC97H with siFARSB. The knockdown efficiency was detected by Western blot (Figure 3A). In CCK-8 and EdU experiments, the cell proliferation level of the FARSB knockdown group was suppressed (Figure 3B,C and Appendix A). Transwell and wound-healing assays confirmed that FARSB knockdown significantly suppressed the migration ability (Figure 3D,E and Appendix A). In addition, we constructed shFARSB stable cell lines and verified the knockdown efficiency of FARSB by Western blot and RT-PCR (Figure 3F). Then, the colony formation assay showed decreased clone numbers in the FARSB knockdown group (Figure 3G and Appendix A). In summary, FARSB can promote HCC progression by promoting proliferation and migration. 

### 2.4. FARSB Activates the mTORC1 Signaling Pathway by Suppressing Raptor Phosphorylation

Several studies have confirmed that ARSs regulate mTORC1 signaling pathways [27,28,29]. Therefore, we verified the relationship between FARSB and mTOR signaling pathway. We used gene set enrichment analysis (GSEA) to perform Kyoto Encyclopedia of Genes and Genomes (KEGG) enrichment analysis and found that the mTOR signaling pathway was significantly enriched in the FARSB high expression group (Figure 4A). The core enrichment molecule contained the core component Raptor of mTORC1 and the core component Rictor of mTORC2 (Appendix A). After siFARSB transfected MHCC97H cells, Western blot showed that the expression level of Raptor decreased after FARSB knockdown, but Rictor did not change significantly, and the activation of mTORC1 was suppressed (Figure 4B). The results showed that rapamycin did not affect the protein expression level of FARSB (Figure 4C and Appendix A), so we speculated that FARSB was the upstream of the mTORC1 complex. At the same time, we constructed FARSB-stable overexpression cell lines and then treated them with 20 nM rapamycin for 48 h. Western blot results showed that rapamycin could block the effect of FARSB gene overexpression on mTORC1 activation (Figure 4D). Then, the co-immunoprecipitation and immunofluorescence co-localization of FARSB and Raptor were performed in MHCC97H cells. The results showed a physical association between FARSB and Raptor (Figure 4E,F). In addition, Western blot showed that the expression level of Raptor decreased, and the phosphorylation level increased when knocking down FARSB, which suggests that knockdown of FARSB reduced the unphosphorylated Raptor but did not increase the amount of phosphorylated Raptor. In contrast, the expression level of Raptor remained unchanged, but the phosphorylation level decreased when overexpressing FARSB (Figure 4G). Previous studies have shown that increased Raptor phosphorylation level suppresses mTORC1 activation [30]. The above results suggest that FARSB activates the mTORC1 signaling pathway by suppressing Raptor phosphorylation.

### 2.5. FARSB Promotes the Proliferation and Migration of HCC Cells by Activating the mTORC1 Signaling Pathway

Subsequently, we studied whether the mTORC1 signaling pathway plays a role in FARSB-related HCC progression. We performed the CCK-8 assay, EdU assay, wound-healing assay and Transwell assay (Figure 5A–D and Appendix A) in MHCC97H cells stably overexpressing FARSB. Rapamycin can block the tumor-promoting effect of FARSB overexpression in HCC, proving that FARSB activates the mTORC1 signaling pathway to promote the proliferation and migration of HCC cells. We further studied the role of FARSB in vivo through a xenograft tumor model. For this reason, MHCC97H cells transfected with FARSB overexpression or FARSB knockdown and control vectors were injected subcutaneously into nude mice. Notably, the weight and volume of tumors in the FARSB knockdown group of mouse xenografts decreased. On the contrary, MHCC97H cells with FARSB overexpression had a faster tumor growth rate than the control group, and rapamycin can block the effect of FARSB on promoting tumor progression. Subsequently, immunohistochemical analysis of the tumor was performed to detect FARSB and proliferating cell nuclear antigen (PCNA). The results showed that FARSB promoted HCC progression in vivo, but rapamycin could block this effect. These results suggest that FARSB promotes HCC cell progressing in vivo and in vitro by activating the mTORC1 signaling pathway (Figure 5E).

### 2.6. FARSB Suppresses Ferroptosis in HCC Cells by Activating mTORC1 Expression 

We further studied the mechanism of FARSB promoting HCC and found that it regulates ferroptosis. We added erastin to MHCC97H cells with knockdown or overexpression of FARSB (with or without rapamycin treatment) and their control groups to induce ferroptosis. We first detected cell viability (Figure 6A) and then ferroptosis core molecule GPX4 (Figure 6B) by Western Blot. Subsequently, ROS (Figure 6C and Appendix A) and Malonydialdehyde (MDA) (Figure 6D), which reflect lipid peroxidation levels, were examined. The results showed that the knockdown of FARSB could enhance erastin-induced ferroptosis but could be blocked by ferroptosis inhibitor ferrostatin-1 rather than apoptosis inhibitor Z-VAD. On the contrary, overexpression of FARSB inhibited ferroptosis, while rapamycin blocked this effect. Finally, the mitochondria of each treatment group were observed under the electron microscope and the length, aspect ratio and fragmentation number of mitochondria were counted, which was consistent with the above results (Figure 6E and Appendix A). In summary, FARSB inhibits ferroptosis in liver cancer cells by activating mTORC1 expression, which may be an essential mechanism for FARSB to promote HCC progression.

## 3. Discussion

Hepatocellular carcinoma (HCC), the main histopathological type of primary liver cancer, is one of the most common causes of cancer death due to its complex etiology and multiple risk factors. However, early screening and diagnosis of HCC are still tricky, so we urgently need to find new effective biomarkers to provide strategies for HCC treatment. Aminoacyl-tRNA synthetases play non-canonical roles other than protein translation in various diseases, including apoptosis, autophagy and ferroptosis, and they are associated with the progression of various tumors [31]. FARSB is a β-regulatory subunit of phenylalanine tRNA synthetase. Its non-canonical role has been reported to be associated with autoimmune and genetic diseases. Bioinformatics analysis data also show it is associated with gastric cancer and lung adenocarcinoma. In the latest study, various bioinformatics analyses found that the hypomethylation of the FARSB promoter in HCC patients led to high expression of FARSB, which was related to the poor prognosis of HCC patients. At the same time, FARSB was associated with immune infiltration and m6A modification. However, the role of FARSB in the progression of HCC has not been experimentally verified.

In this study, we performed bioinformatics analysis using data from the TCGA database to show that FARSB is highly expressed in HCC patients and is associated with a poor prognosis of patients, consistent with previous research results. The experimental data we provided showed that the expression level of FARSB was up-regulated in the tissues of HCC patients and HCC cell lines in vitro. Next, we confirmed the role of FARSB in the progression of HCC for the first time. By knocking down the expression of FRASB in HCC cell lines, we found that FARSB is closely related to the proliferation and migration of HCC cells, but the effect of FARSB on the invasion of HCC cells has not been confirmed.

Several studies have confirmed that ARS is involved in regulating the mTORC1 signaling pathway. For example, leucyl-tRNA synthetase (LeuRS) plays a classic role in protein translation and acts as an intracellular leucine sensor to mediate mTORC1 activation [32]. Isoleucyl-tRNA synthetase 2 (IleRS2) affects the proliferation of lung cancer cells by regulating the mTOR pathway [21]. The mTORC1 signaling pathway controls cell growth, proliferation and metabolism. It plays a vital role in the progression of HCC.mTORC1, a crucial nutritional sensing complex, and its target is the core of the cell metabolic sensing network. Knockdown of tRNA synthetase will lead to an amino acid deprivation reaction, which belongs to the amino acid metabolic pathway imbalance [33]. Then, it is likely related to the mTORC1 signaling pathway. Therefore, we performed GSEA enrichment analysis on FARSB and found that the mTOR signaling pathway was significantly enriched in the FARSB high expression group. After knocking down FARSB, it was found that the activation level of mTORC1 decreased, and the suppression effect of rapamycin on the growth of HCC cells could be blocked by FARSB overexpression, so we confirmed that FARSB regulates the progression of HCC through the mTORC1 signaling pathway. Although the association between FARSB and the mTORC1 signaling pathway was verified, the relationship between FARSB and other signaling pathways remains to be studied.

We found a physical association between FARSB and Raptor through co-immunoprecipitation and immunofluorescence co-localization experiments. Therefore, we speculate that the interaction between FARSB and Raptor suppresses its phosphorylation and activates the mTORC1 signaling pathway to promote HCC progression. However, this study only verified the endogenous interaction between FARSB and Raptor, and the exogenous interaction still needs to be verified. In addition, the interaction between FARSB and Raptor and the specific regulatory mechanism of FARSB on Raptor phosphorylation have yet to be carried out in detail in this study.

FARSB knockdown enhanced the sensitivity of erastin to ferroptosis in HCC cells, while overexpression of FARSB showed the opposite effect. We also further verified that FARSB regulates ferroptosis through the mTORC1 signaling pathway, which may be another mechanism by which FARSB affects HCC progression. It has been reported that FARSB is a downstream target molecule of transcription factor 4 (ATF4), and the inhibition of tRNA synthetase will lead to the phosphorylation of elF2α and the activation of ATF4 to enhance the expression of aminoacyl-tRNA synthetase under amino acid starvation [4]. Therefore, the decrease in FARSB expression will lead to the up-regulation of ATF4 expression, and studies have shown that ATF4 can suppress the expression of GPX4 [34], which may be the specific mechanism of FARSB regulating ferroptosis. Since the ferroptosis inducer erastin suppresses the expression of cystine glutamate reverse transporter (xCT) in the $X_{c}^{-}$ system, we detect the expression level of GPX4 via Western blot. It may be necessary to use GPX4 inhibitor (1S,3R)-RSL3(RSL3) to detect further whether the expression level of xCT changes after FARSB knockdown. In addition, whether FARSB knockdown can enhance the sensitivity of HCC treatment drugs with ferroptosis as the primary mechanism of HCC cells has not been expanded.

Overall, our study provides multidimensional evidence for FARSB as a potential biomarker and prognostic factor for HCC. Our results show that the high expression of FARSB in HCC is closely related to patients’ low survival and poor prognosis. FARSB promotes HCC progression by suppressing Raptor phosphorylation and activating the mTORC1 signaling pathway. At the same time, FARSB suppresses ferroptosis through the mTORC1 signaling pathway, which may also promote HCC progression.

## 4. Material and Methods

### 4.1. Bioinformatics Analysis 

Transcriptome sequencing data and clinical data of patients with hepatocellular carcinoma were obtained from the TCGA database “https://portal.gdc.cancer.gov/ (accessed on 5 November 2022)”. Four hundred fifty-five samples were obtained, including sixty normal and three hundred ninety-five tumor tissues. R software 4.2.1 was used for differential expression analysis, univariate and multivariate Cox regression analysis, forest plot and survival analysis. Kegg enrichment analysis was performed using GSEA software “http://www.gsea-msigdb.org/gsea/index.jsp (accessed on 6 November 2022)”. 

### 4.2. Patients and Tissue Specimens 

Liver cancer patients (*n* = 65) who underwent hepatectomy in the Department of Hepatobiliary Surgery, Union Hospital, Tongji Medical College, Huazhong University of Science and Technology from October 2020 to April 2022 were collected (Appendix A). The Ethics Committee of Union Hospital approved all procedures, and written informed consent was obtained from all patients before surgery. Each sample was divided into two parts: one was used to make tissue microarrays and the other was stored.

### 4.3. Immunohistochemistry 

Human HCC tissue microarrays were dewaxed with xylene and hydrated with ethanol gradient. The blocked sections were incubated with FARSB (16341-1-AP, 1: 1000 dilution, Proteintech, Wuhan, China) antibody. The sections of subcutaneous tumors in nude mice were incubated with anti-FARSB (16341-1-AP, 1:1000 dilution, Proteintech, Wuhan, China) and PCNA (10205-2-AP, 1:2000 dilution, Proteintech, Wuhan, China). The sections were then incubated with HRP-conjugated secondary antibodies and stained with 3,3′-diaminobenzidine (DAB). Two independent pathologists scored all IHC samples. The IHC score was divided into staining intensity score (0: negative, 1: weakly positive, 2: moderately positive and 3: strongly positive) and staining positive area score (0: 0%, 1: 10–25%, 2: 26–50%, 3: 51–75% and 4: 76–100%).

### 4.4. Cell Culture 

LO2, MHCC97H, Huh7 and HepG2 cells were purchased from Procell (Wuhan, China) and confirmed by STR analysis. All experiments used cells with less than 20 passages; mycoplasma detection was not performed on cell lines. Culture cells according to the recommended guidelines. The cells were cultured in basal medium containing 1% penicillin/streptomycin (BOSTER, Wuhan, China), 10% fetal bovine serum (HYCEZMBIO, Wuhan, China) and Dulbecco’s modified Eagle’s medium (DMEM) (GIBCO, Grand Island, NY, USA) at 37 °C (containing 5% CO_2_). 

In order to inhibit mTOR, ferroptosis and apoptosis, cells were treated with rapamycin (S1039, 20 nM, Selleck, Houston, TX, USA), ferrostatin-1 (S7243, 2 μM, Selleck, Houston, TX, USA) and Z-VAD-FMK (S7023, 10 μM, Selleck, Houston, TX, USA) to induce ferroptosis. Cells were treated with Erastin (S7242, 10 μM, Selleck, Houston, TX, USA).

### 4.5. Immunofluorescence 

Immunofluorescence staining of LO2, MHCC97H, Huh7 and HepG2 cells was performed. HCC cells were fixed on cell slides with 4% paraformaldehyde-closure at room temperature (goat serum) for 1 h. The blocked HCC cells were incubated with primary antibodies of FARSB (16341-1-AP, 1: 200 dilution, Proteintech, Wuhan, China) and Raptor (20984-1-AP, 1: 200 dilution, Proteintech, Wuhan, China) at 4 °C overnight. CY3-conjugated donkey anti-rabbit IgG (GB21403, diluted 1:100, Servicebio, Wuhan, China) and FITC-conjugated affiniPure goat anti-rabbit IgG (BA1105, diluted 1:100, Boster, Wuhan, China) were used. The nuclei were stained with Hoechst (BL803A, Biosharp, Hefei, China). The anti-fluorescence quencher was added, and the slides were observed under a microscope (Olympus CX31 microscope, Tokyo, Japan).

### 4.6. Cell Transfection and Lentivirus Infection 

siRNA targeting FARSB was purchased from Genecreate (Wuhan, China) and transfected into cells using lipo8000 (Beyotime, Shanghai, China). Plasmids containing shRNA targeting FARSB were purchased from Tsingke (Wuhan, China), plasmids encoding full-length FARSB were purchased from Genchem (Shanghai, China) and lentiviral packaging plasmids psPAX2 and pMD2G were purchased from Tsingke (Wuhan, China). The target plasmid and the lentiviral packaging plasmid were co-transfected into HEK293T cells using lipo8000. After 48 h of transfection, the supernatant was collected and filtered with a 0.22 μm filter to obtain the virus solution, which was added to the medium of Huh7 and MHCC97H together with polybrene (Beyotime, Shanghai, China) to select cells with stable puromycin resistance. The sequences of shRNA and siRNA are listed in the Appendix A.

### 4.7. Reverse Transcription and RT-PCR 

Total RNA was extracted using TRIzol reagent (Takara, Tokyo, Japan) and reverse transcribed using PrimeScript RT (Takara, Tokyo, Japan). RT-qPCR analysis was performed using ChamQ SYBR qPCR premix (Vazyme, Nanjing, China) and a real-time fluorescence quantitative PCR detector (Bio-Rad, Hercules, CA, USA). β-actin was used as an endogenous control to compare CT (2^−ΔΔCT)^. Primer sequences are listed in the Appendix A.

### 4.8. Western Blot

The cells were lysed with RIPA buffer containing protease inhibitor and phosphatase inhibitor (MedChemExpress, Monmouth Junction, NJ, USA) to extract proteins and then treated with ultrasound. The exact amount of protein samples was separated by SDS-PAGE and transferred to the PVDF membrane. After the skim milk was blocked, the PVDF membrane was incubated overnight with the corresponding primary antibody at 4 °C. The PVDF membrane was then rinsed three times and incubated with the corresponding HRP-conjugated AffiniPure goat anti-rabbit (mouse) IgG (GB23303, GB23301, 1:3000 dilution, Servicebio, Wuhan, China) at room temperature for one hour. Subsequently, these proteins were visualized and semi-quantitatively analyzed using the ChemiDoc imaging system (Bio-Rad, Hercules, CA, USA). Antibody: FARSB (16341-1-AP, 1:1500 dilution, Proteintech, Wuhan, China), β-actin (60008-1-Ig, 1:5000 dilution, Proteintech, Wuhan, China), GAPDH (10494-1-AP, 1:5000 dilution, Proteintech, Wuhan, China) and Raptor (20984-1-AP, 1:1000 dilution, Proteintech, Wuhan, China). Rictor (27248-1-AP, 1:1000 dilution, Proteintech, Wuhan, China), p70S6K (# 2708, 1:1000 dilution, Cell Signaling Technology, Danvers, MA, USA), p-p70S6K (# 9206,1: 1000 dilution, Cell Signaling Technology, Danvers, MA, USA), 4e-BP1 (# 9452, 1:1000 dilution, Cell Signaling Technology, Danvers, MA, USA), p-4e-BP1 (# 9456, 1:1000 dilution, Cell Signaling Technology, Danvers, MA, USA), p-Raptor (# 2083, 1:1000 dilution, Cell Signaling Technology, Danvers, MA, USA) and GPX4 (T56959, 1:1000 dilution, abmart, Shanghai, China).

### 4.9. Cell Proliferation Assay 

Two thousand cells were inoculated into 96-well plates (100 μL DMEM per well) and incubated in the dark for 30 min according to the steps of the CCK8 kit of Beyotime (Shanghai, China). The absorbance at 450 nm was measured with a Multiskan TM GO microplate spectrophotometer (Thermo Fisher Scientific, Waltham, MA, USA). Next, follow the manufacturer’s instructions to operate the BeyoClick TM EdU-488 Cell Proliferation Assay Kit (Beyotime, Shanghai, China) and then observe the 96-well plate under the microscope (Olympus CX31 microscope, Tokyo, Japan).

### 4.10. Wound-Healing Assay 

The cells were fostered in a 6-well plate to a single-layer fully covered well and cultured in DMEM without FBS for 24 h. The cells were scraped with a 200 μL pipette tip to create a wound, and the distance between the scratch edges of each treatment group was tested at a specified time. 

### 4.11. Transwell Assay

The 24-well Corning cell culture chamber with 8μm pore size was used. The cancer cells were suspended in 200 μL DMEM without FBS and inoculated into the upper chamber, and 600 μL 20% FBS DMEM was added to the lower chamber. The cells were fixed with 4% paraformaldehyde for 30 min at room temperature and stained with crystal violet. The microscope shot four random points in each chamber and counted them.

### 4.12. Colony Formation Assay 

Two thousand cells were inoculated in 6-well plates and cultured for 7 days. Then, it was fixed with 4% paraformaldehyde at room temperature for 30 min and stained with crystal violet. 

### 4.13. Co-IP 

The protein was extracted from HCC cells with 0.4 ml RIPA buffer, and the protein sample was incubated with 30 μL protein A + G agarose beads (Beyotime, Shanghai, China) at 4 °C for one hour. After centrifugation, the supernatant was transferred to a new centrifuge tube. The portion was taken as an input sample, and then the corresponding primary antibody was added and incubated at 4 °C overnight. Subsequently, 30 μL protein A + G agarose beads were added and incubated at 4 °C for two hours. After centrifugation, the supernatant was removed. The obtained agarose beads were washed with PBS 6 times, and then a 2X SDS loading buffer was added. The solution was treated in a metal bath at 95 °C for 10 min, and the supernatant was used for Western blotting.

### 4.14. Xenograft Tumor Model in Nude Mice 

MHCC97H cells (1 × 10^6^) infected with FARSB knockdown lentivirus, overexpressing lentivirus, and their respective control viruses were inoculated subcutaneously into the right flank of nude mice (BALB/c nu, 6 weeks, male). Six mice in each FARSB knockdown and control group were sacrificed 20 days later, and the tumors were removed and weighed. The overexpression group and the control group had 12 mice in each group. After 8 days, the two groups of mice were divided into two groups on average. Half of the two treatment groups were injected with normal saline and half with rapamycin (4 mg/kg). After 16 days, the mice were sacrificed, and the tumors were removed and weighed. The tumor volume was measured every four days (V = 0.5 × L × W^2^, where L is the length and W is the width). After obtaining tumor specimens, the tumors were sectioned and the IHC was analyzed.

### 4.15. ROS Detection 

Intracellular ROS levels were measured using a fluorescent probe 2′,7′-dichlorofluorescein diacetate (DCFH-DA, Cat # S0033S, Beyotime, Shanghai, China). The slides inoculated with the cells of each treatment group were incubated in an Opti-MEM medium with the DCFH-DA probe at 37 °C for 30 min. The anti-fluorescence quencher was added, and the slides were observed under a microscope (Olympus CX31 microscope, Tokyo, Japan). 

### 4.16. MDA Detection 

The treated cells were digested with trypsin, washed with pre-cooled PBS solution and lysed with an ultrasonic cell disruptor. After centrifugation at 1500× *g* at 4 °C for 10 min, the cell debris was removed and the supernatant was collected. The MDA content was detected by ELISA according to the steps of the MDA kit (E-EL-0060c, Elabscience, Wuhan, China).

### 4.17. Mitochondrial Electron Microscopy 

Sample preparation: 10 cm dishes with cell density of 80–90%; without rinsing, the medium was quickly poured out, and a sufficient amount of 4 °C pre-cooled electron microscope special fixative was added for one hour. The fixed cells were quickly shoveled, transferred to a centrifuge tube with a cell scraper, and centrifuged at 800 rpm for five minutes until agglomerated. Most of the supernatant was lightly aspirated and transferred to a new centrifuge tube to extend the natural sedimentation time as required. The supernatant was discarded, and a new electron microscope fixative was added to preserve it in a refrigerator at 4 °C.

### 4.18. Statistical Analysis 

Statistical analysis was performed using GraphPad Prism 9.5 software. The differences between the two groups were analyzed via unpaired *t*-test, and the data between multiple groups were analyzed via one-way ANOVA. Multiple comparisons were performed using two-way ANOVA. The experimental data were obtained by three or more independent repeated experiments, expressed as mean ± SD; when *p* < 0.05, the difference is considered significant.

## 5. Conclusions

FARSB promotes HCC progression and is associated with a poor prognosis of patients. FARSB activates the mTORC1 signaling pathway by binding to the component Raptor of the mTORC1 complex and plays a role in promoting cancer. 

FARSB is expected to be a biomarker for early screening and treatment of HCC.

## Figures and Tables

**Figure 1 ijms-24-16709-f001:**
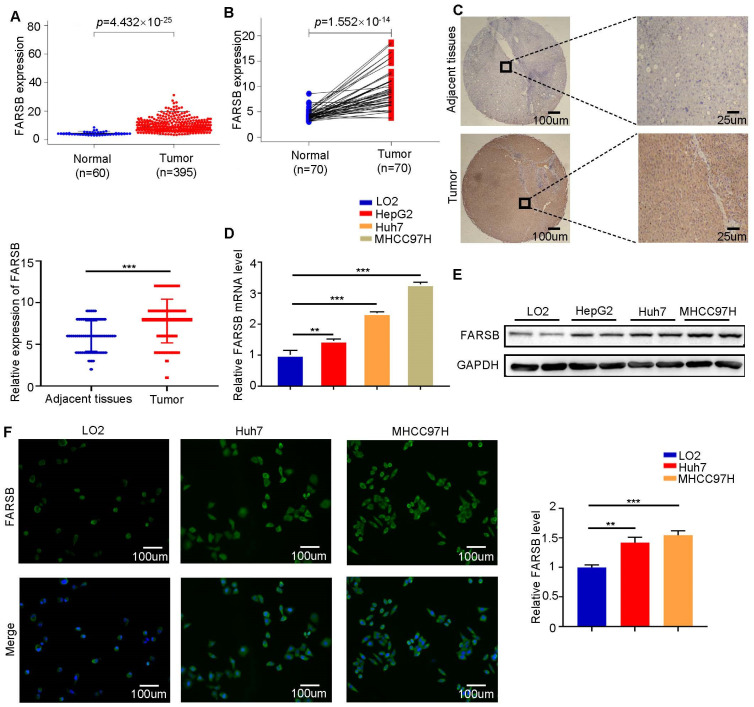
FARSB is highly expressed in hepatocellular carcinoma. (**A**,**B**) The mRNA level difference of FARSB between tumor and normal tissues in the TCGA-LIHC dataset (**A**) and the mRNA level difference of FARSB between tumor and adjacent tissues (**B**). (**C**) Representative results of IHC and IHC score (*n* = 65). Scale bars: 100 μm (insets 25 μm) (**D**,**E**) RT-PCR (**D**) and Western blot analysis (**E**) of normal liver cell lines and liver tumor cell lines. (**F**) FARSB immunofluorescence analysis of LO2, Huh7 and MHCC97H. Scale bars: 100 μm, **: *p* < 0.01, ***: *p* < 0.001.

**Figure 2 ijms-24-16709-f002:**
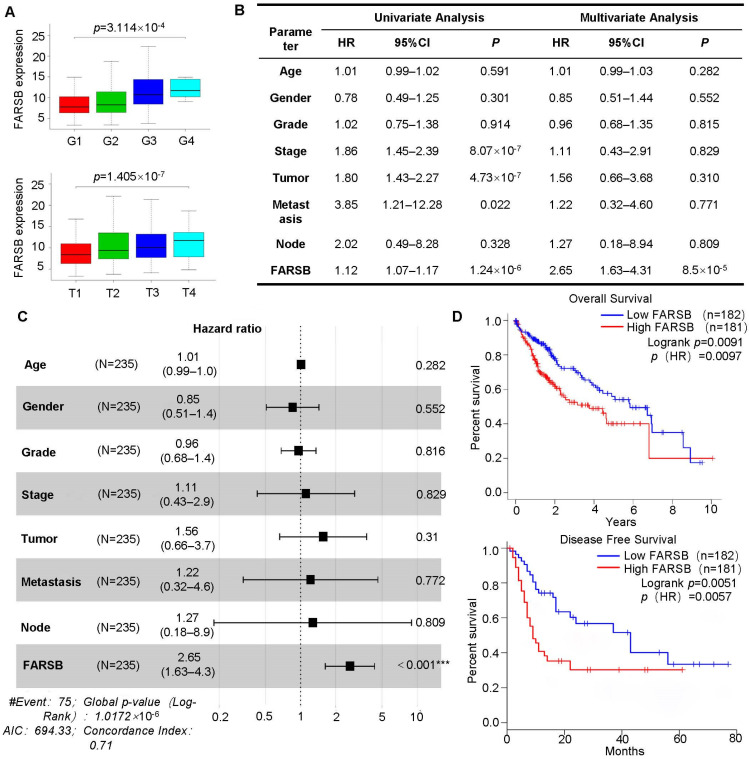
A high expression of FARSB is associated with a poor prognosis of patients. (**A**) The correlation between FARSB mRNA level, tumor grade and TMN stage in the TCGA-LIHC dataset. In the (**B**,**C**) TCGA-LIHC dataset, Cox regression analysis was performed on the clinical correlation data, and the forest plot was drawn with the data in the table (*n* = 235). (**D**) The TCGA-LIHC dataset was used for patient survival analysis, including overall survival and disease-free survival. ***: *p* < 0.001.

**Figure 3 ijms-24-16709-f003:**
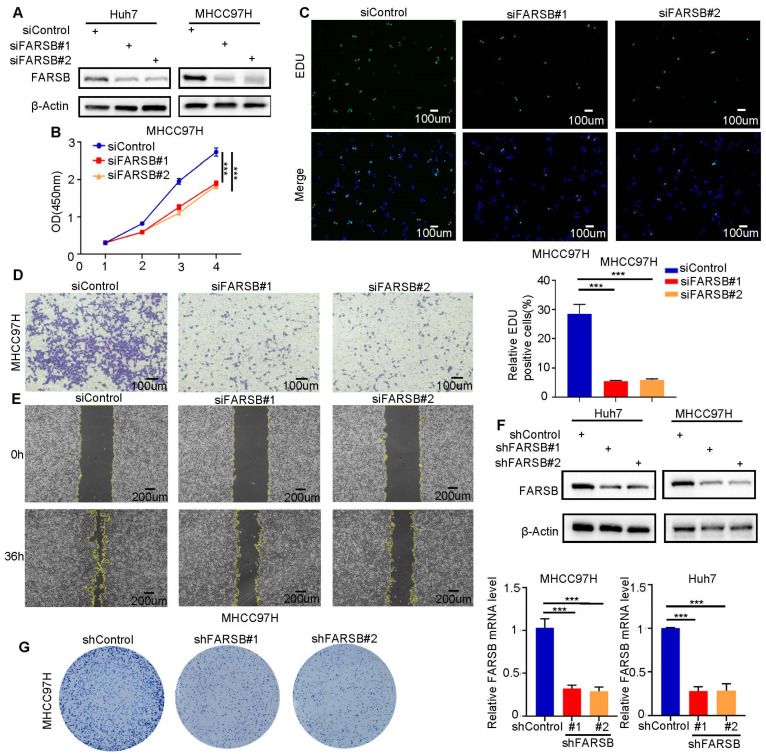
FARSB knockdown suppresses the proliferation and migration of HCC cells. (**A**) Western blot analysis of FARSB protein levels in Huh7 and MHCC97H cells in knockdown and control groups. (**B**,**C**) CCK-8 (**B**) and EdU (**C**) showed that knockdown of FARSB suppressed the proliferation of MHCC97H. Fluorescent images showing EdU positive cells nuclei (green) and total nuclei in the samples (Hoechst, blue). Scale: 100 μm. (**D**,**E**) The Transwell (**D**) and scratch test (**E**) showed that knockdown of FARSB suppressed the cell migration of MHCC97H cells. Scale: 100 μm. (**F**,**G**) The FARSB knockdown-stable cell line was constructed, and its knockdown efficiency (**F**) was tested by Western blot and RT-PCR. The colony formation assay showed that the colony formation ability of FARSB knockdown MHCC97H cells decreased (**G**). ***: *p* < 0.001.

**Figure 4 ijms-24-16709-f004:**
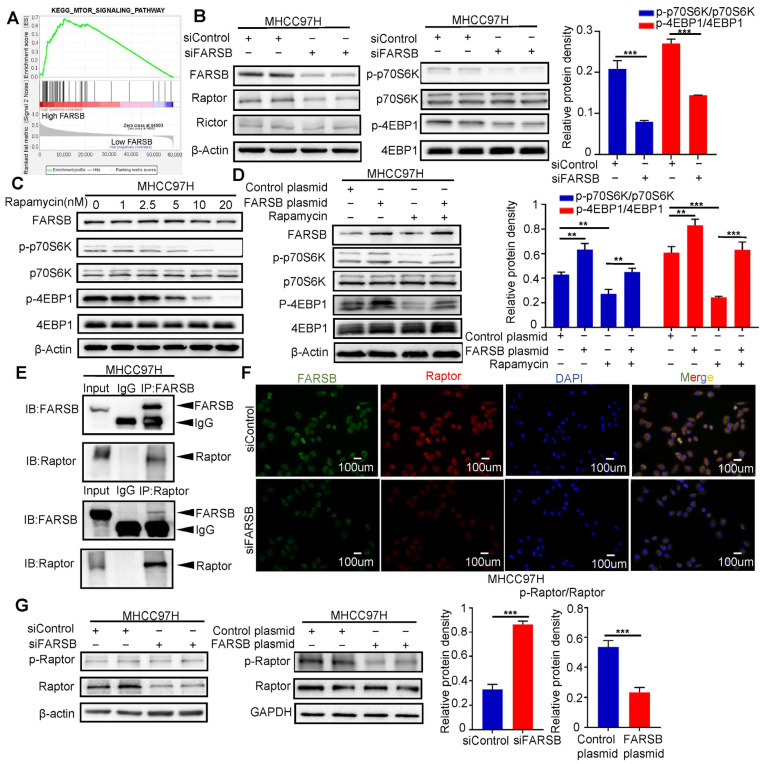
FARSB activates the mTORC1 signaling pathway by suppressing Raptor phosphorylation. (**A**) Using the TCGA-LIHC dataset, GSEA performed Kegg enrichment analysis, and the mTOR signaling pathway was significantly enriched in the FARSB overexpression group. (**B**) Western blot detected the activation level of mTORC1 after FARSB knockdown. (**C**) MHCC97H cells were treated with rapamycin at a gradient concentration for 24 h, and Western blot detected the expression level of FARSB and the activation level of mTORC1. (**D**) MHCC97H cells overexpressing FARSB and the control group were treated with 20 nM rapamycin for 48 h, and Western blot detected the activation level of mTORC1. (**E**,**F**) Co-immunoprecipitation assay (**E**) and immunofluorescence co-localization (**F**) confirmed the endogenous (MHCC97H) interaction between FARSB and Raptor. (**G**) Western blot detected the changes in the phosphorylation level of Raptor after knockdown or overexpression of FARSB. **: *p* < 0.01, ***: *p* < 0.001.

**Figure 5 ijms-24-16709-f005:**
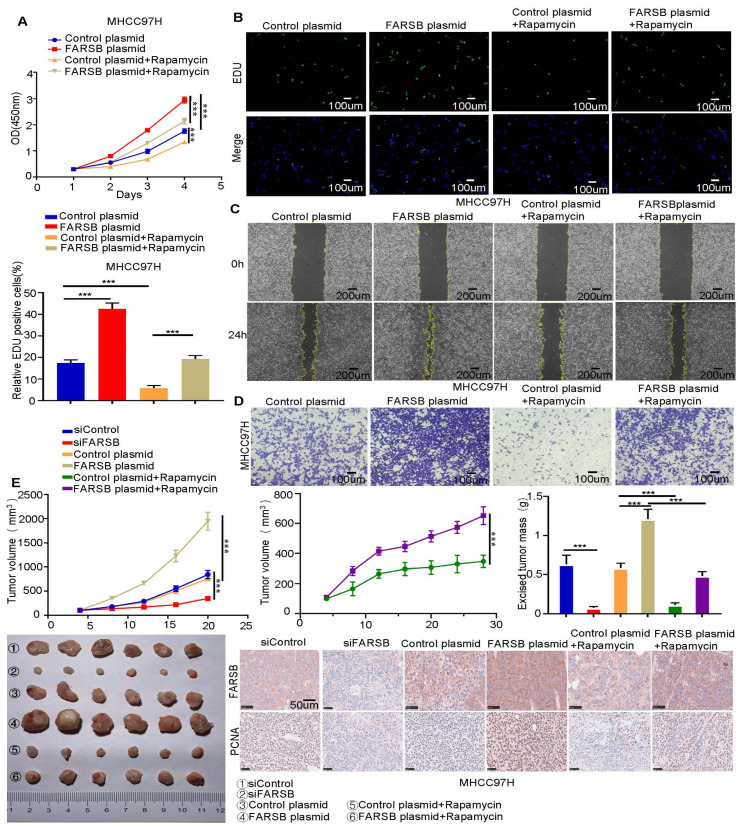
FARSB promotes the proliferation and migration of HCC cells by activating the mTORC1 signaling pathway. (**A**,**D**) Overexpression of FARSB and control group MHCC97H cells were treated with 20 nM rapamycin for 48 h. CCK-8 (**A**) and EdU (**B**) were used to detect the proliferation of MHCC97H cells. Fluorescent images showing EdU positive cells nuclei (green) and total nuclei in the samples (Hoechst, blue). Wound-healing assay (**C**) and Transwell assay (**D**) were used to detect the migration ability of MHCC97H cells. (**E**) Knockdown or overexpression of FARSB and control group MHCC97H cells were used for the xenograft tumor model in nude mice. Among them, FARSB overexpression and control group mice began on the eighth day; half of the nude mice were injected with normal saline and half were injected with rapamycin (4 mg/kg) for 16 days. The changes in tumor volume were recorded at different time points, and the subcutaneous tumors were removed and weighed. Representative immunohistochemical images showed the expression of FARSB and PCNA in xenograft tumor tissues. Scale: 50 μm, ***: *p* < 0.001.

**Figure 6 ijms-24-16709-f006:**
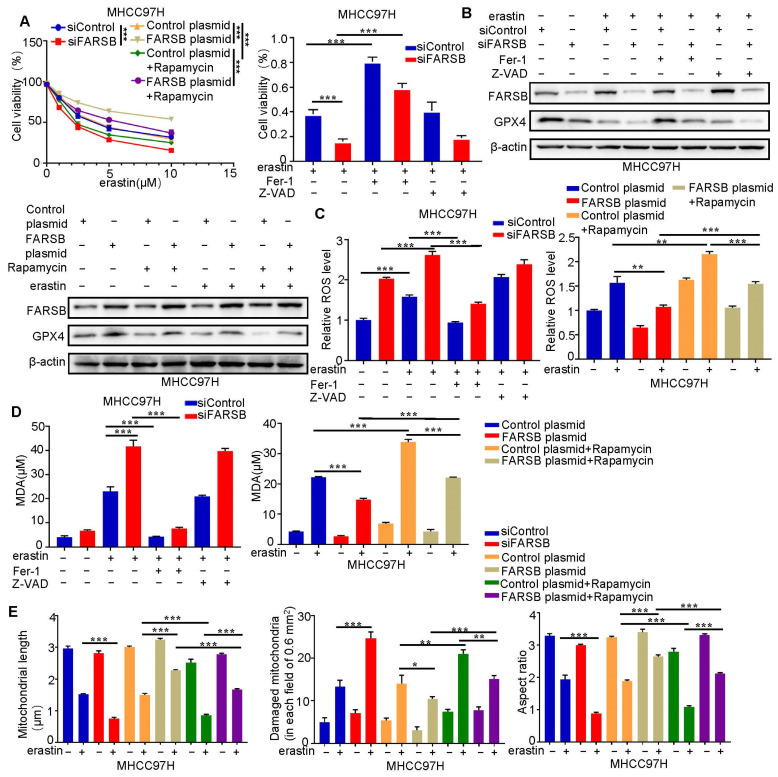
FARSB suppresses ferroptosis in HCC cells by activating mTORC1 expression. (**A**) Knockdown or overexpression of FARSB (with or without 20 nM rapamycin) and MHCC97H cells in the control group were treated with gradient concentrations (0, 1, 2.5, 5, 10) of erastin to detect cell viability. FARSB knockdown MHCC97H and control cells were treated with 10 μM erastin and then ferrostatin-1 (2 μM) or Z-VAD (10 μM) for 24 h, respectively. Cell viability (**B**–**D**) was measured. FARSB knockdown or overexpression MHCC97H cells were treated with 10 μM erastin, with or without ferrostatin-1 (2 μM), Z-VAD (10 μM) and rapamycin (20 nM), respectively. The expression level of GPX4 was detected via Western blot (**B**). The ROS (**C**) and MDA (**D**) levels in each treatment group were detected. (**E**) The 10 μM erastin-treated MHCC97H cells with knockdown or overexpression of FARSB and MHCC97H cells with overexpression of FARSB (with or without 20 nM rapamycin). The morphology of mitochondria in each group was observed under the electron microscope, and the length of mitochondria, the number of mitochondrial damage and the aspect ratio were counted. *: *p* < 0.05, **: *p* < 0.01, ***: *p* < 0.001.

## Data Availability

All data used in this study were from open access websites described in materials and methods.

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
