# Peer review of "FARSB Facilitates Hepatocellular Carcinoma Progression by Activating the mTORC1 Signaling Pathway"

_ijms, 2023, doi:10.3390/ijms242316709_

Round 1
Reviewer 1 Report
Comments and Suggestions for Authors
In this article, the authors showed that FARSB is highly expressed in hepatocellular carcinoma from the TCGA database and that the expression of FARSB is associated with poor prognosis of patients suffering hepatocellular carcinoma. Then, the authors analyzed the FARSB-mediated signaling pathway in a hepatocellular carcinoma cell line and demonstrated that FARSB can interact with Raptor, a subunit of mTOR complex. FARSB can decrease the repressing phosphorylation of Raptor leading to the activation of mTORC1. Then, the authors demonstrated that FARSB can promote cell proliferation, migration and ferroptosis in mTORC1-dependent manner.
I found the study to be is well done and the results are convincing. I only suggest some minor modifications.
Line 92, Figure 1: Could authors describe the cell lines used. I assumed that one of them is a untransformed cell line. From the Figure 1F, it seems to me that FARSB is a nuclear protein in LO2 cells and mainly cytoplasmic in Huh7 and MHCC97H. Is that right? Is it of importance? If yes, could authors discuss this point.
Line 108: could authors precise T stage. Is that stage based on the TMN classification?
Figure 3B: the authors used CCK8 kit to analyze cell proliferation. Could authors explain CCK8.
Figure 3C: It is difficult to detect EDU on my computer. Could authors increase the light or contrast.
Line 152: could authors precise that Rapamycin is a mTOR inhibitor.
Figure 6G and line 160: I can not detect the increase in p-Raptor. However, I agree that the ratio p-Raptor/total raptor is increased. This suggest that silencing of RARSB decreased the unphosphorylated form of Raptor but did not enhanced the amount of phosphorylated Raptor.
Line 164: Could authors moderate their claim and indicate that “the above results suggest” instead of indicate”.
Line 210: could authors precise the target or mechanisms of action of erastin.
Author Response
Dear Reviewers,
Thank you very much for your time involved in reviewing the manuscript and your very encouraging comments on the merits.
Comments: In this article, the authors showed that FARSB is highly expressed in hepatocellular carcinoma from the TCGA database and that the expression of FARSB is associated with poor prognosis of patients suffering hepatocellular carcinoma. Then, the authors analyzed the FARSB-mediated signaling pathway in a hepatocellular carcinoma cell line and demonstrated that FARSB can interact with Raptor, a subunit of mTOR complex. FARSB can decrease the repressing phosphorylation of Raptor leading to the activation of mTORC1. Then, the authors demonstrated that FARSB can promote cell proliferation, migration and ferroptosis in mTORC1-dependent manner. I found the study to be is well done and the results are convincing. I only suggest some minor modifications.
Response: We also appreciate your clear and detailed feedback and hope that the explanation has fully addressed all of your concerns. In the remainder of this letter, we discuss each of your comments individually along with our corresponding responses. To facilitate this discussion, we first retype your comments in italic font and then present our responses to the comments.
Comment 1: Line 92, Figure 1: Could authors describe the cell lines used. I assumed that one of them is a untransformed cell line. From the Figure 1F, it seems to me that FARSB is a nuclear protein in LO2 cells and mainly cytoplasmic in Huh7 and MHCC97H. Is that right? Is it of importance? If yes, could authors discuss this point.
Response 1: We gratefully appreciate for your valuable suggestion.
(1) The HCC cell lines used in this study were Huh7, MHCC97H and HepG2, and the normal liver cell line LO2, which were purchased from Procell ( Wuhan, China ) and confirmed by STR analysis. According to the literature, 1 Huh7 is derived from the liver of a 57-year-old Japanese male with liver cancer. It is reported to produce alpha-fetoprotein, trypsin alpha antibody, plasma ceruloplasmin, fibrinogen, and fibronectin and is an epithelial cell-like adherent cell. MHCC97H cells are derived from the MHCC97 cell line of a 39-year-old male patient with liver cancer. MHCC97H cells are highly metastatic sublines which inherit maternal HBV gene-positive cell characteristics and Y chromosome loss. HepG2 cells were isolated from hepatocellular carcinoma of a 15-year-old white male youth with liver cancer. The cells expressed genes such as alpha-fetoprotein, β-lipoprotein, and retinol-binding protein. There is no evidence that the hepatitis B virus genome exists in HepG2 cells. In addition to the so-called hepatocellular carcinoma or hepatocellular carcinoma cell lines based on the original publication[1], HepG2 is also known as hepatoblastoma cell line[2] and is an epithelial cell-like adherent cell. The L-02 cell population doubling time is about 20 hours, the cell morphology is epithelioid cells, the cell proliferation is rapid, the expression of AFP and CK-19 is negative, the expression of ALB is at the level of 10μg / ml, some cells undergo morphological changes after multiple passages, and the expression of ALB is missing.
(2)It is reported that the charging of tRNA molecules with phenylalanine in the cytoplasm is via a tetrameric enzyme that contains two catalytic subunits encoded by phenylalanyl-tRNA synthetase alpha ( FARSA ) and two regulatory subunits encoded by phenylalanyl-tRNA synthetase beta ( FARSB )[3]. From the present cellular immunofluorescence results, the subcellular localization of FARSB is in the cytoplasm, and some nuclei have fluorescence. The following reasons are considered: ①The incubation time of primary antibody and secondary antibody is extended, resulting in nuclear false positive. ② If the image is not taken at the optimal z-axis level, there may be nuclear-cytoplasmic overlap, resulting in false positive nuclear fluorescence. However, it is not excluded that in normal cells and tumour cells, the difference in the expression of FARSB in the nucleus and cytoplasm needs further proof. In addition, we found other FARSB immunofluorescence results in LO2 cells in another non-representative figure, which may better illustrate the subcellular localization of FARSB in the cytoplasm.
Comment 2:Line 108: could authors precise T stage. Is that stage based on the TMN classification?
Response 2: We apologize for this article's lack of specific expression. T classification is a tumour classification method based on the TMN staging system to describe the primary tumour. T1 ~ T4 expresses t classification with increased tumour volume and adjacent tissue involvement. According to patients' clinical information from the TCGA database, T1: tumour diameter ≤ 2 cm or tumour> 2 cm did not spread to blood vessels. T2: Tumor > 2 cm, spread to blood vessels or multiple tumours; the tumour is 5 cm. T4: A single tumour or multiple tumours of any size that spread to the main branch of the portal vein or hepatic vein or spread to nearby organs ( except the gallbladder ) or tumours that invade the peritoneum.
Comment 3:Figure 3B: the authors used CCK8 kit to analyze cell proliferation. Could authors explain CCK8.
Response 3: The CCK8 Cell Counting Kit-8 reagent is used for simple and accurate cell proliferation and toxicity analysis. The basic principle is that the reagent contains water-soluble tetrazolium salt WST-8 [ chemical name: 2- ( 2-methoxy-4-nitrophenyl ) -3- ( 4-nitrophenyl ) -5- ( 2,4-disulfobenzene ) -2H-tetrazole monosodium salt ], which is reduced by dehydrogenase in cells to form a highly water-soluble yellow formazan dye under the action of the electron carrier 1-methoxy-5-methylphenazine dimethyl sulfate ( 1-Methoxy PMS ). The amount of formed formazan is proportional to the number of living cells. The faster the cell proliferation, the darker the colour; the more significant the cytotoxicity, the lighter the colour. There is a linear relationship between the colour depth and the number of cells for the same cells, so that this feature can be used directly for cell proliferation and toxicity analysis.
Comment 4:Figure 3C: It is difficult to detect EDU on my computer. Could authors increase the light or contrast.
Response 4: We apologize for the lack of light and contrast in the EDU image in this article. We have modified the original EDU image ( including Figure 3C, Figure S1B, Figure 5B, Figure S3B ).
Figure3C
FigureS1B
Figure5B FigureS3B
Comment 4: Line 152: could authors precise that Rapamycin is a mTOR inhibitor.
Response 4: Rapamycin, also known as sirolimus, is the first known mTOR inhibitor initially isolated from Streptomyces hygroscopicus. It was approved by the U.S. Food and Drug Administration ( FDA ) in 1999 as an immunosuppressant for preventing organ transplant rejection[4]. Rapamycin inhibits the signal transduction pathway required for cell growth and proliferation by forming a functional complex with peptidyl-proline-isomerase FK-506 binding protein 12 ( FKBP12 )[5]. Rapamycin binds to mTORC1 at the C-terminus by interacting with the immunoaffinity protein FKBP12. Immunoaffinity protein FKBP12 is a non-specific mTOR interacting protein that binds to rapamycin only by binding to the mTORC1 complex[6]. Rapamycin binds to FKBP12 and acts explicitly as an allosteric inhibitor of mTORC1. In this study, we used rapamycin to inhibit the mTORC1 signaling pathway. After treatment with 20 nM rapamycin, the phosphorylation levels of its downstream marker molecules p70S6K and 4eBP1 were detected by WB, which suggests the suppression of mmTORC1.
Comment 5: Figure 6G and line 160: I can not detect the increase in p-Raptor. However, I agree that the ratio p-Raptor/total raptor is increased. This suggest that silencing of FARSB decreased the unphosphorylated form of Raptor but did not enhanced the amount of phosphorylated Raptor.
Response 5: Many thanks to your kind reminder. Sorry for our lack of rigorous expression, we modify line 160 as 'In addition, Western Blot showed that the expression level of Raptor decreased, and the phosphorylation level increased when knocking down FARSB,which suggests that knockdown of FARSB reduced the unphosphorylated Raptor but did not increase the amount of phosphorylated Raptor. In contrast, the expression level of Raptor re-mained unchanged, but the phosphorylation level decreased when overexpressing FARSB. '
Comment 6:Line 164: Could authors moderate their claim and indicate that “the above results suggest” instead of “indicate”.
Response 6: We appreciate your valuable suggestions. We have changed ' the above results indicate ' in lines 96 and 166 to ' the above results suggest '.
Comment 7:Line 210: could authors precise the target or mechanisms of action of erastin.
Response 7: Erastin is a ferroptosis activator. Erastin can mediate ferroptosis through various molecules, including
- Cysteine-glutamate transporter receptor ( system): System is a reverse transporter protein in the plasma membrane. Glutamate was transferred from the cells, and cystine was transferred to the cells at a ratio of 1 : 1. Erastin can prevent extracellular cystine from entering cells by inhibiting the system , reducing intracellular glutathione levels. Glutathione is an indispensable substrate for the antioxidant effect of GPX4. Therefore, if the activity of GPX4 is reduced, the redox homeostasis will be decomposed, and L-ROS will accumulate, leading to oxidative cell death, that is, ferroptosis[7].
- Voltage-dependent anion channel ( VDAC ): The regulation of VDAC opening by erastin will significantly affect mitochondrial metabolism. This will increase oxidative phosphorylation and ROS production and cause oxidative stress, eventually leading to ferroptosis[8].
③ p53: erastin Activation of p53 may play an essential role in tumour suppression by inhibiting SLC7A11 transcription and ultimately inhibiting ferroptosis[9].
[1] Knowles BB, Howe CC, Aden DP. Human hepatocellular carcinoma cell lines secrete the major plasma proteins and hepatitis B surface antigen. Science. 1980 Jul 25;209(4455):497-9
[2] López-Terrada D, Cheung SW, Finegold MJ, Knowles BB. Hep G2 is a hepatoblastoma-derived cell line. Hum Pathol. 2009 Oct;40(10):1512-5
[3] Rodova M, Ankilova V, Safro MG. Human phenylalanyl-tRNA synthetase: cloning, characterization of the deduced amino acid sequences in terms of the structural domains and coordinately regulated expression of the alpha and beta subunits in chronic myeloid leukemia cells. Biochem Biophys Res Commun. 1999 Feb 24;255(3):765-73
[4] Mao B, Zhang Q, Ma L, Zhao DS, Zhao P, Yan P. Overview of Research into mTOR Inhibitors. Molecules. 2022 Aug 19;27(16):5295
[5] Jayaraman T, Marks AR. Rapamycin-FKBP12 blocks proliferation, induces differentiation, and inhibits cdc2 kinase activity in a myogenic cell line. J Biol Chem. 1993 Dec 5;268(34):25385-8
[6] Lipton JO, Sahin M. The neurology of mTOR. Neuron. 2014 Oct 22;84(2):275-91
[7] Skouta R, Dixon SJ, Wang J, Dunn DE, Orman M, Shimada K, Rosenberg PA, Lo DC, Weinberg JM, Linkermann A, Stockwell BR. Ferrostatins inhibit oxidative lipid damage and cell death in diverse disease models. J Am Chem Soc. 2014 Mar 26;136(12):4551-6
[8] Lemasters JJ. Evolution of Voltage-Dependent Anion Channel Function: From Molecular Sieve to Governator to Actuator of Ferroptosis. Front Oncol. 2017 Dec 19;7:303
[9] Wang SJ, Li D, Ou Y, Jiang L, Chen Y, Zhao Y, Gu W. Acetylation Is Crucial for p53-Mediated Ferroptosis and Tumor Suppression. Cell Rep. 2016 Oct 4;17(2):36

Reviewer 2 Report
Comments and Suggestions for Authors
This study provides multidimensional evidence for FARSB as a potential biomarker and prognostic factor for HCC. Our results show that the high expression ofFARSB in HCC is closely related to patients' low survival and poor prognosis. FARSB promotes HCC progression by suppressing Raptor phosphorylation and activating the mTORC1 signaling pathway. At the same time, FARSB suppresses ferroptosis through the mTORC1 signaling pathway, which may also promote HCC progression.
This is a very well written and interesting article. I have some comments.
1. A more detailed explanation is needed as to why the authors targeted FARSB. What is the strength of the expression with other mTOR-related factors?
2. What are the patient backgrounds of the HCC samples used in this paper? Are there any changes in FARSB expression between HBV, HCV, and non-viral HCC?
3. Is direct sampling from cancer required for FARSB analysis? Is it possible to analyze blood samples? In figure 2, the details are unclear as to which part of the tissue is used, so please explain.
4. Is there a significant difference in figure 2D Kaplan-Meier? Since there seems to be a strong correlation between FARSB expression and T factor, is it possible that T factor is correlated with prognosis?
5.What do the red and blue bars in figures S2 and S4 represent? There is no explanation.
Author Response
Dear Reviewers,
Thank you very much for your time involved in reviewing the manuscript and your very encouraging comments on the merits.
Comments: This study provides multidimensional evidence for FARSB as a potential biomarker and prognostic factor for HCC. Our results show that the high expression of FARSB in HCC is closely related to patients' low survival and poor prognosis. FARSB promotes HCC progression by suppressing Raptor phosphorylation and activating the mTORC1 signaling pathway. At the same time, FARSB suppresses ferroptosis through the mTORC1 signaling pathway, which may also promote HCC progression. This is a very well written and interesting article. I have some comments.
Response: We also appreciate your clear and detailed feedback and hope that the explanation has fully addressed all of your concerns. In the remainder of this letter, we discuss each of your comments individually along with our corresponding responses. To facilitate this discussion, we first retype your comments in italic font and then present our responses to the comments.
Comment 1: A more detailed explanation is needed as to why the authors targeted FARSB. What is the strength of the expression with other mTOR-related factors?
Response 1: It has been mentioned in the text that the biallelic mutation of FARSB can lead to cirrhosis[1] ( PMID: 30014610 ), which is one of the leading causes of hepatocellular carcinoma. In another study of metabolic disorders in gastric cancer, metabolomics analysis of 16 gastric cancer tissues showed that the aminoacyl-tRNA biosynthesis pathway in gastric tissues was significantly up-regulated compared with adjacent non-cancerous tissues, and the expression level of phenylalanine-tRNA synthetase ( FARSB ) was associated with tumour grade and low survival rate, respectively. In addition, gastric tissue array data analysis showed that FARSB was up-regulated in gastric cancer tissues and was associated with poor prognosis and tumour metastasis. Therefore, we intend to explore the role of FARSB in HCC further [2]. In another study, a more comprehensive bioinformatics analysis of FARSB was performed. According to the TIMER database, it was found that the expression level of FARSB was significantly up-regulated in HCC and many other tumours. The data set LIRI-JP downloaded from ICGC, GSE76427 downloaded from GEO, and HPA-based data analysis confirmed that the expression level of FARSB in liver tumour tissues of HCC patients was significantly higher than that in normal tissues[3]. However, this study only analyzed FARSB from the bioinformatics perspective, so we further explored the role of FARSB in the experiment.
In this study, we only studied the mTORC1-related factors and also excluded the role of the mTORC2 component Rictor. However, other mTOR-related components have yet to be further explored, and further experiments are needed to verify them.
Comment 2: What are the patient backgrounds of the HCC samples used in this paper? Are there any changes in FARSB expression between HBV, HCV, and non-viral HCC?
Response 2: We apologise for not providing detailed information about liver cancer patients in the tissue microarray. Here is a statistical table of background information of patients, in which patients were excluded from HBV and HCV infection. There is no record of HBV and HCV infection in the information of liver cancer patients with TCGA, and the expression of FARSB in liver cancer patients with HBV and/or HCV still needs further study.
Table S1 Baseline characteristics of patients whose samples were used in the tissue microarray
|
Clinical information |
N(%) |
|
No. of patients |
65 |
|
Age ,yr |
59.2±10.3 |
|
Gender (male/female) |
42/21 |
|
Histology |
Hepatocellular carcinoma |
|
Stage |
|
|
Ⅰ |
5 (7.7%) |
|
Ⅱ |
15 (23.1%) |
|
Ⅲ |
33 (50.8%) |
|
Ⅳ |
12 (18.5%) |
|
T stage |
|
|
T1 |
3 (4.6%) |
|
T2 |
13 (20%) |
|
T3 |
38 (58.5%) |
|
T4 |
11 (16.9%) |
|
Lymph nodes |
|
|
N0 |
20 (30.8%) |
|
N1 |
12 (18.5%) |
|
N2 |
18 (27.7%) |
|
N3 |
15 (23.1%) |
|
Metastasis |
|
|
M0 |
52 (80%) |
|
M1 |
13 ((20%) |
|
Grade |
|
|
G2 |
13 (20%) |
|
G2-3 |
14 (21.5%) |
|
G3 |
31 (47.7%) |
|
G3-4 |
7 (10.8%) |
Comment 3: Is direct sampling from cancer required for FARSB analysis? Is it possible to analyze blood samples? In figure 2, the details are unclear as to which part of the tissue is used, so please explain.
Response 3: We apologize for not elaborating on the samples used in this study. The samples of patients used in the TCGA database and the tissue microarray we produced were from HCC tumours and their normal adjacent tissues. FARSB is a cytoplasmic rather than a secretory protein and generally does not exist in the blood, so we do not use blood samples for research.
Comment 4: Is there a significant difference in figure 2D Kaplan-Meier? Since there seems to be a strong correlation between FARSB expression and T factor, is it possible that T factor is correlated with prognosis?
Response 4:We gratefully appreciate for your valuable suggestion. We have supplemented the relevant information to the original picture.
We did not conduct in-depth research on T classification. In a previous study, logistics analysis showed that T classification was associated with prognosis[3].
Comment 5: What do the red and blue bars in figures S2 and S4 represent? There is no explanation.
Response 5: Sorry for the lack of markup; we added relevant information to the text.
FigureS2
FigureS4
[1] Zadjali F, Al-Yahyaee A, Al-Nabhani M, Al-Mubaihsi S, Gujjar A, Raniga S, Al-Maawali A. Homozygosity for FARSB mutation leads to Phe-tRNA synthetase-related disease of growth restriction, brain calcification, and interstitial lung disease. Hum Mutat. 2018 Oct;39(10):1355-1359
[2] Gao X, Guo R, Li Y, Kang G, Wu Y, Cheng J, Jia J, Wang W, Li Z, Wang A, Xu H, Jia Y, Li Y, Qi X, Wei Z, Wei C. Contribution of upregulated aminoacyl-tRNA biosynthesis to metabolic dysregulation in gastric cancer. J Gastroenterol Hepatol. 2021 Nov;36(11):3113-3126
[3] Zhen J, Pan J, Zhou X, Yu Z, Jiang Y, Gong Y, Ding Y, Liu Y, Guo L. FARSB serves as a novel hypomethylated and immune cell infiltration related prognostic biomarker in hepatocellular carcinoma. Aging (Albany NY). 2023 Apr 3;15(8):2937-2969
